# Assessing the Value of Multimodal Interfaces: A Study on Human–Machine Interaction in Weld Inspection Workstations

**DOI:** 10.3390/s23115043

**Published:** 2023-05-24

**Authors:** Paul Chojecki, Dominykas Strazdas, David Przewozny, Niklas Gard, Detlef Runde, Niklas Hoerner, Ayoub Al-Hamadi, Peter Eisert, Sebastian Bosse

**Affiliations:** 1Fraunhofer HHI, 10587 Berlin, Germany; david.przewozny@hhi.fraunhofer.de (D.P.); niklas.gard@hhi.fraunhofer.de (N.G.); detlef.runde@hhi.fraunhofer.de (D.R.); niklas.hoerner@hhi.fraunhofer.de (N.H.); peter.eisert@hhi.fraunhofer.de (P.E.); sebastian.bosse@hhi.fraunhofer.de (S.B.); 2Neuro-Information Technology, Otto-von-Guericke-University Magdeburg, 39106 Magdeburg, Germany; dominykas.strazdas@ovgu.de (D.S.); ayoub.al-hamadi@ovgu.de (A.A.-H.)

**Keywords:** human–machine interaction, multimodal interface, human–robot interaction, spatial computing

## Abstract

Multimodal user interfaces promise natural and intuitive human–machine interactions. However, is the extra effort for the development of a complex multisensor system justified, or can users also be satisfied with only one input modality? This study investigates interactions in an industrial weld inspection workstation. Three unimodal interfaces, including spatial interaction with buttons augmented on a workpiece or a worktable, and speech commands, were tested individually and in a multimodal combination. Within the unimodal conditions, users preferred the augmented worktable, but overall, the interindividual usage of all input technologies in the multimodal condition was ranked best. Our findings indicate that the implementation and the use of multiple input modalities is valuable and that it is difficult to predict the usability of individual input modalities for complex systems.

## 1. Introduction

Since humans and robots act in a three-dimensional physical world, it is a natural approach to design human–robot interactions (HRIs) spatially as well, especially if tasks are related to 3D positions. To enable such collaborations not only for expert users, these interactions also need to be natural and intuitive. Thus, traditional two-dimensional interfaces, e.g., the mouse and keyboard are not perfectly suitable for such tasks. Multimodal user interfaces can provide better solutions, as they combine various interaction modalities to enable more flexibility [1] and naturalness [2] in human–machine interactions (HMI). They allow the seamless adaptation to the user and application requirements and enable higher usability, acceptance and efficiency. Multimodal HMIs have already been successfully implemented in various domains, e.g., for sterile medical device controls [3] and appear to be beneficial in complex interactions between humans and collaborative robots (cobot), e.g., to support human competence in assembly scenarios [4].

Based on the requirements of an industrial use case, we developed an HRI concept for a user-friendly human–robot welding seams inspection workstation. Our general objectives are to increase ergonomics and comfort during the inspection of a welded frame, allow the simple and fast spatial documentation of the inspection results, and to control the robot and inspection process with ease. Our solution is designed around the idea that the robot presents the workpiece to the user at a personalized body height, rotates the workpiece if needed and picks the next workpiece after the prior one has been inspected. In contrast to the currently used solution, where users have to bend over the workpiece, which is fixed in an unchangeable position, in our solution, they can stand in an ergonomically favorable position and are intuitively assisted in documentation. To avoid distracting the user from the inspection task, we designed an easy-to-use interface, close to the inspection area. This interface for documentation and robot control should be, as in many industrial interfaces, also usable with working gloves and with one hand holding a tool.

We designed and implemented several interaction concepts to meet all the requirements mentioned above. We evaluated these different interaction modalities in a novel multisensory system (see Section 3) composed of the following components: a collaborative robot arm, 6-degree-of-freedom (DOF) vision-based object tracking, augmented reality visualization, facial and gesture recognition, speech input/output, and a projection-based interface. Furthermore, we evaluated the reliable operation of our hand tracking and speech input solutions in technology-specific pre-tests (see Section 4).

In accordance with the guidelines for multimodal interaction design [1], we implemented an augmented, touch-able interface and speech input and output. The augmented interface is projected in two alternative areas (workpiece and worktable) and activated by touch events, recognized via computer vision. The interface projected on the workpiece completely overlaps the inspection area in front of the user, and thus prevents unnecessary head movements. However, to interact with this interface, users must raise their arms to at least chest height. The interface projected onto the worktable in front of the user, has the disadvantage of additional downward head movement, out of the inspection focus, but allows for more convenient hand interactions at the hip level. Users of speech interface can also stay visually focused within the inspection area, but need to memorize voice commands. Thus, although specifically designed for this spatial HRI task, each interaction modality has its advantages and drawbacks. However, it is not clear, whether a parallel, multimodal usage of these interactions might be beneficial for spatial HRI tasks.

In this paper, we report a user evaluation (see Section 5) of the documentation task in our use case. Here, we compare four interaction modalities: touch interaction with an augmented interface on (1) the worktable or (2) on the curved workpiece, (3) a voice control and (4) a multimodal interaction combining all three unimodal methods. Our results (see Section 6) show that test subjects prefer the multimodal interaction. Surprisingly, voice control was often used within the multimodal condition, although it has the lowest user ranking within the unimodal conditions. The interaction on the worktable, and not on the inspected workpiece, is the best unimodal condition.

## 2. Related Work

Multimodal interaction has been studied for decades. Sharon Oviatt has identified a collection of “10 myths” that are held as widely accepted wisdom—unfortunately, they may not be true [2]. A particularly interesting myth in the context of this paper is “Myth #4: Speech is the primary input mode in any multimodal system that includes it”.

Multimodal interfaces pose specific challenges, such as data fusion and real-time processing [5,6], but reward the effort with better stability [7] and reliability [8]. In addition, compared to unimodal interfaces [9], a higher user acceptance is achieved without compromising the process through a higher perceived mental workload [4,10]; an aspect that is beneficial for ergonomic and health reasons. Furthermore, the redundancy of input options allows for higher flexibility in adapting to changing conditions [1].

In a review paper about industrial HRI [11], Berg and Lu pointed out: “There is a need for the combination of existing approaches of interfaces, like gestures, speech, or even touch displays”. A recent survey in HRI was performed in [12]. They systematized the vast and large field of unimodal input and output modes used in HRI. Multimodal combinations of these modalities are not part of this consideration. Recent surveys of multimodal input [13,14] include surveys of human–robot collaboration (HRC). All these surveys cover different aspects at the intersection of human–computer interaction (HCI) and robotics.

In this work, we did not consider the safety aspects of the HRI. The current state of how to ensure this important issue can be found in [15]. To ensure a smooth HRI situational zone-based robot control should be implemented as shown by Bdiwi et al. [16].

### Related Systems

There are numerous systems evaluating different input modalities in HRI. Some of them perform Wizard of Oz (WoZ) experiments to prevent the imperfect recognition of gesture or speech inputs, whilst others risk the evaluation of implemented live systems. Maternet et al. [17] found that up to 10% of errors of a recognition system can be tolerated when testing a multimodal system. A comparison between touch, speech and a combination of both [18] showed that “participants strongly prefer multimodal input to unimodal input”. It was noted that some participants were frustrated by speech recognition.

Projection-based augmentation was used by Zhou et al. [19] in industrial welding scenarios to assist in the inspection process. They referred to the new technology as spatial augmented reality (SAR), wherein a projection highlights the welds to be examined and guides the user step by step. In this approach, the augmentation information comes from a digital 3D model of the workpiece, which must be manually calibrated, and the user input comes from a five-button controller. In this approach, the focus is on introducing SAR technology, while features such as automatic calibration, workpiece tracking, and multimodal user input are missing.

In another multimodal setup combining speech and gesture interaction, users preferred the only speech condition [10]. The system was evaluated using NASA TLX [20] and the System Usability Scale (SUS) [21]. Abich and Barber [10] reported that SUS results become more significant if users rate the different modalities after completing all of them.

Differences in user preference and acceptance for input modalities in similar HRI scenarios were discovered in a few WoZ studies, showing that users are faster when using gestures and 6D pointing instead of using touch [17] and prefer a combination of pointing gestures and descriptive voice commands [22]. Both studies stated that test subjects preferred to use hand gestures, even though the recognition system contained forced flaws.

Rupprecht et al. [23] projected an assembly instruction onto an industrial workpiece and compared two different approaches to providing information, “static in-view instructions” (always at a central position) and “guided in situ instructions” (at changing positions, in close proximity to the next assembly position). Interaction was realized via two different gestures, one for confirming task completion and continuing the workflow (forward-function), and one for returning to the last instruction (backward-function).

## 3. System Design

This section describes the use case and its workflow, as well as the technical implementation with the modules used.

### 3.1. Use Case

Visual weld inspection, as common in industrial production or energy industry, is a demanding procedure which requires specialized training, attention, significant physical effort [24].

For our study, we chose the subframe, the heavy structural component of a car, as the workpiece. In practice, it is typically inspected by manually rotating the workpiece in a mount around an axis and visually checked by the worker for defective weld from an uncomfortable body posture (see Figure 1, left). Small defects, such as protruding burrs, are immediately corrected, while in the case of larger defects, the workpiece is either sent for repair or sorted out. There is no documentation of the defects. To streamline the system, workpieces should be presented in an optimal position to the worker; documentation should be performed directly on the inspection site so that additional time consumption is minimal.

Our solution approach is based on the idea that the workpiece is presented to the worker by a robot in any position and orientation convenient to the worker, respecting their body dimensions, in contrast to the current inspection set-up, where the workpiece is fixed in one position and the worker has to adapt (see Figure 1, right). For robot control, defect detection and marking, and documentation, we implemented a multimodal interaction approach, i.e., combining hand and voice inputs, and a projective interface on a worktable or on the workpiece itself, leaving it up to the user to choose their preferred interaction mode for each step. For the development of multimodal user interfaces, we used the guidelines in [1].

### 3.2. Workflow

The inspection process (see Figure 2) starts with the robot fetching a new workpiece and presenting it to the inspector for visual examination. If the inspector does not find a defect during this view, the robot is instructed to move the workpiece to another position. This is performed, as in the following steps, either by speech or by touching an augmented/projected button or interaction area. If a defect is found, the next step is to mark its position by having the operator point to the corresponding location on the workpiece. The next step is to classify the type of defect. After that, a decision is made regarding whether to repair the defect on site, whether to send the workpiece for advanced repair or it is not repairable and should be sorted out. Once the decision is confirmed, the workpiece is released and the process starts over with a new subframe. Hand interaction can take place on the table underneath the workpiece as well as on the workpiece itself. If errors occur, or safety measures prohibit robot movement, a message is initiated by a speech module and a text-to-speech (TTS) engine.

### 3.3. Setup

In addition to the robot handling of the metal frame, the setup consists of cameras, microphones, projectors, and computers. Software are highly modularized and communicate with one another via a middleware (see Section 3.4).

The projection and registration module (see Section 3.8) uses a projector–camera system (industrial camera and laser hybrid projector Casio XJ-A257) for the projective augmentation and registration of scene objects [25]. It translates hand positions to object space and communicates with the web application to proceed in the workflow. To interact with the augmented interface, a hand detection module (see Section 3.5) is employed. It detects and tracks the hand and finger positions, allowing the interaction with projected buttons and icons by pointing or touching. The speech recognition module (see Section 3.7) provides spoken hints and detects speech commands. To ensure that only intended speech commands are executed, sound direction is evaluated as well as the information of the head module (see Section 3.6), which detects and recognizes faces in front of the workspace. To cover a wide interaction range, two Microsoft Kinect 2 cameras are necessary. Due to the driver restriction of a single Kinect per computer, each hand-tracking software has to run on its own PC.

For calibration between the Kinect/top-view cameras, the robot, and the projector, we relied on chessboard detection and World/Robot-Tool-Flange calibration with 3D-EasyCalib™ [26]. All modules are explained in detail in the following sections.

### 3.4. Middleware

To integrate the required components with many external hardware connections (cameras) and a high computational load, a distributed software architecture is advisable (see Figure 3).

OPC UA [27] or robot operating system (ROSs) [28] are widely used systems that connect such components into distributed environments [29]. However, each of these systems have some drawbacks regarding its interfaces and pre-processing options that make it tedious to build an interaction system based on them. Therefore, we implemented a small but flexible middleware tailored to our application. To create a device-independent user interface (UI), we decided for web technologies. Based on this decision, we selected Websockets to transfer real-time data instead of a webservice, as bidirectional communication is possible here.

Instead of using a simple message broker, some processing is performed in the middleware. Each sensor and application can use its own coordinate system specific to its camera or display. All coordinate transformations are handled inside the middleware. Additional data filtering, sensor fusion and system-wide state calculations are also performed herein.

### 3.5. Hand Recognition Module

This module enables two types of bare hand interaction. Near the workpiece, we have a projective interface with touch and in addition, we enable contact-free hand tracking and gesture recognition.

Our mid-air interaction is based on the Microsoft Kinect 2 skeleton tracking. Starting from the Kinect hand positions, the point cloud of a 2cm×20cm×20cm cube from the depth image is extracted, and the arm is truncated at the estimated wrist position. The grayscale 2D representation of the point cloud is calculated and further improved by a morphological filter, and the hand position is calculated by using the center of gravity. Using a convolutional neural network (CNN), the hand gesture is inferred. The selected grayscale depth image is scaled to 128 × 128 pixels and converted into a histogram of gradient orientations (HOG). The HOG is processed by a net of three convolutional and pooling layers and two fully connected layers.

The touch-based hand interaction is based on foreground-background segmentation. Due to the projective interface, the segmentation of the foreground and background is difficult on color images. Therefore, depth and infrared images from the Kinect camera are used. By reason of resolution and depth errors near depth discontinuities, the depth image alone is not sufficient, while in the infrared image the contrast between the hand and background may also be marginal. After post-processing by morphological filters and segmentation, the fusion of the two difference images is sufficient for hand detection. In the segmented hand image, the fingertip is detected. The position of the fingertip, the position of the orthogonal projection of the fingertip on the background and the fingertip–background distance are calculated.

As in the case of the air-only interface, the hand pose is computed by the aforementioned CNN. If the robot has moved and no motion is detected, the background image is updated in the depth and infrared channels and the segmentation is recalculated.

### 3.6. Head Orientation Module

This face recognition module was implemented according to Saxen et al. [30] and is based on the YOLO architecture (yolov3-tiny) [31] trained on the ImageNet dataset [32]. The transposed YOLO architecture for face detection, using transfer learning and adapted loss function, is used to predict a face direction vector (FDV), which can be used to calculate a rotated bounding box around a face. This approach proved to work better than alternative face detection systems, capable only calculating ordinary bounding boxes [30]. For head pose estimation, a simple landmark-based method was explored (called head pose on top of facial landmarks, or HPFL), which gave significantly better results when compared to different landmark detection methods, which is also capable of estimating the head pose [30]. Compared to the head pose estimation provided by OpenFace [33], this method provides a significantly better head pose estimation.

### 3.7. Speech Modules

Three submodules concerning speech were developed and used in our setup:Text-to-speech: We implemented a standard browser-based (HTML5 and Javascript) application for text-to-speech conversion (Web Speech API, no Internet connection required). This application receives InstructionMessages from other components and outputs the received messages (e.g., hints to the worker) via a loudspeaker.Speech recognition: Another web application was connected to a microphone and uses the Web Speech API [34] within a Chrome browser (Internet connection required) for online speech recognition. The recognized text is then checked for stored commands. These commands are linked to certain instructions, so that it is possible to control the whole workflow by speech. For each command, several phrases of correct commands and similar sounding variations are acceptedSound direction detection: The Microsoft Kinect 2 has a built-in microphone array, which can be used to detect the direction of sound. Thus, we implemented a C#-application as a middleware client, which reads the sound direction from the Kinect sensor and streams these data to the interaction module. This module can then use this information to determine whether the received voice command originates from the person facing the workpiece or from somewhere else.To reject speech commands from users other than the operator and from the operator not momentarily operating (i.e., facing) the system, we reject recognized speech commands when the user is not looking at the workpiece, i.e., the head orientation deviates by more than 20° from a straight view towards the workpiece or the audio beam (received from a microphone array behind the workpiece) deviates by more than 30° sideways from the working position (i.e., someone else is speaking).

### 3.8. Projection and Registration Module

The projection and registration module visually links the work environment, consisting of the workpiece, a worktable, and the robot system to the user. It is aware of the relative 3D position of the scene elements and performs projection mapping to present information that matches the local environment. The appearance of objects with known 3D geometry is manipulated by rendering a synthetic image of the objects from the viewpoint of the projector [35]. The projector forms a calibrated unit with a camera that tracks the 6D pose of the objects [36], as shown in Figure 4.

A contour-based tracking approach combines an edge-correspondence optimization and a dense refinement for real-time 6D registration [37]. An approximate initial workpiece pose is known from the robot end-effector that grabs the object. Nevertheless, image-based pose refinement is necessary to achieve accurately fitting augmentations even with fluctuating grasp positions. To further optimize projection matching accuracy, an initial calibration refinement optimizes the extrinsic parameters of the projector–camera system given the precisely known pose of a 3D object [25]. A depth buffer-based mask removes interfering seams at the object border by applying a morphological erosion on the object silhouette [38]. Figure 5 shows the appearance of a highlighted weld using the real workpiece. We integrated three different types of spatial augmentation into our system.

Augmentation of welds: Each weld is modeled as a separate texture layer using the texture coordinates of the 3D model (Figure 6, right). Each layer is a binary image containing a transparent background and the exact path of the weld. Middleware commands can control the projected color. The module transforms the incoming hand positions into object coordinates and triggers interactions with the weld if the hand is within a spherical area surrounding it. If the user’s hand moves only briefly through the sphere, the weld flashes, awaiting confirmation by resting for a short time.User interface on a workpiece: We directly projected a menu structure on the workpiece surface to enhance the interaction flow. Depending on the number of buttons needed, the 3D model is sliced vertically. For each slice, an axis-aligned bounding box represents the interaction region (Figure 6, left). Hand interaction is handled as described above. All texture pixels in that box are found by projecting back to the texture space. The respective texture layer is filled with a randomly sized and oriented pictograph. The variation in scale and orientation ensures the readability of the pictograph from different positions and distances.User interface on worktable: A web-based interface is projected on the worktable. The projection is pre-distorted to create a rectangular image on the surface. The off-screen renderer renders the website into the texture of a flat 3D object and updates it at high frequency. The perpendicular projection of the center of the hand into the projection surface creates a cursor position in 3D space. It is converted into 2D pixel coordinates and a metric height.

### 3.9. Robot Module

The robot module is responsible for the control of the robot. It activates the sub-programs necessary for picking up, displaying, or putting down the workpiece. The robot routine consists of pre-programmed positions, that are required to view the workpiece from all directions in order to check the critical and mandatory welds. During the inspection process, the robot can be put into a “free-drive” mode, allowing the robot-assisted movement of the workpiece.

In this mode, the user perceives the workpiece as weightless, as the weight is being compensated by the robot, still allowing movement, as the robot still reacts to additional perceptual forces and moment effects. The current position of the robot tool center point, and thus of the workpiece, is being transmitted via the middleware to the projection module for an easier initial guess of the orientation.

For the initial setup of the system, a 3D-printed lightweight and robust polylactic acid (PLA) mockup of the workpiece was used, as it allows safe interaction and the use of smaller robots. The mockup was further enhanced with an additional structure for easier and precise gripping.

The mockup and original metal workpieces showed both good results for object tracking and augmentation.

## 4. Technical Assessment of Recognition Performance

To make our system more comparable, we give some short performance measurements of our user input.

### 4.1. Performance Middleware

It is crucial for a real-time interaction system to sustain low latency, independently of the communication layer or the workload. We compared our own middleware with a mosquitto MQTT [39] broker (v. 1.5.8) as an example of a standard middleware. On a PC (Xeon E5-2650 v3, 32 GB RAM, Windows 10), we ran a message sender, the middleware, and as a client, a web page in a Chrome browser (v. 96) or a middleware client written in C#.

For different sender frequencies, we measured the latency from message generation to message recognition in the web page or client. When using our middleware, latency was a little higher with the web client than with MQTT, and a little lower with the C# client (see Figure 7). The additional system load for the sender, middleware, and browser was negligible. The frequency of messages typically used in our system did not have any influence on the latency.

### 4.2. Speech Recognition

Our restriction on voice command recognition (see above) may reduce the recognition rate. We tested the recognition rate of simple voice commands and the stability of head orientation and sound direction detection. In a noisy laboratory (44 dB background noise from a server), we tested the recognition rate at two positions and with two head orientations.

The “work position” is 1 m apart from the workpiece, camera, and microphone. The test subjects looked at the workpiece and the camera. Half a meter laterally to the workpiece, speech commands and actions were displayed to the test person. To test the head orientation, the test subject had to remain in the work position and look to the side so that the head orientation deviated by more than 20°.

To test the sound direction, the test person had to move 1.5 m sideways from the working position. This is outside the camera’s field of view, and no head direction is detected here. Voice commands were accepted if the sound direction was no more than 30° off axis. The test persons had to adapt to the different fixed postures and speak a randomly selected phrase. We tested 13 subjects (11♂, 2♀), who spoke 33 phrases in each of the three postures. The experiments were conducted in German language in December 2021 using the German version of Google’s online speech recognition.

In the working position, the speech command recognition rate was 82%; the head orientation had no effect. At the lateral position, the recognition rate was 64%. At the workplace, we have a nearly perfect 99.7% accuracy in audio beam detection. Only one speech command was rejected due to incorrect direction. In 7% of the voice commands spoken at the side position, working positions were assigned due to the incorrect beam direction. Furthermore, 6% of the speech commands were rejected due to incorrectly detected head orientation.

When the subject is standing at the work position and looking sideways, the system incorrectly estimates in 23% of the cases that the user is looking at the workpiece. We did not investigate whether this shortcoming was caused by detection errors or whether the test subjects were actually looking in the wrong direction. By displaying the test task on a monitor near the workpiece, it was not always easy to remember to read the task, turn the head, and then start to speak the voice command.

Our system largely excluded unintended voice commands by accepting only those from speakers who were in front of the workpiece and facing it. The recognition rate decreased by only 6% through this process. Up to 92% of speech commands, where the speaker was not in the correct pose, were rejected. However, there are other possibilities for recognizing voice commands, such as when a person in a working position and a person to the side are speaking at the same time, which have yet to be tested.

### 4.3. Hand Detection Accuracy

In the course of the development of the bird’s-eye view hand tracking, an evaluation of the measurement accuracy was carried out by moving an artificial hand on a measurement table in the detection range of the hand tracking in parallel to the table level of the system (see Figure 8). The tracking values transformed into the table level, as also used for interaction, resulting in an approximately linear measurement, especially for the height values that are difficult to capture (see Figure 9).

The standard deviations of the measurements are quite small under the overall conditions, averaging at approximately 12 mm across all measurements. For our use-case, this accuracy is more than sufficient, since our interactive elements and the inspected welds are significantly larger.

## 5. Uni- vs. Multimodal Interaction Evaluation

To shift the focus on the interaction and not to add unintended effects from the robot, we use a reduced and stationary study setup, which also allows the use of the original steel workpiece.

### 5.1. Study Design

In our user trial, we wanted to test whether our system was suitable for the inspection task and evaluate the performance of different interaction modalities. As an objective measurement, we determine the task completion time, and for the user experience, the NASA Task Load Index (NASA TLX) [20,40] (in German, see [41]), the System Usability Score (SUS) [21,42] (in German, see [43]), a short version of User Experience Questionnaire (UEQ-S) [44] and user satisfaction with Kunin scale [45]. This study used a within-subject design so that all subjects experienced all modalities.

Test subjects had to mark a defective weld, tag it with an error type, enter the handling mode, confirm the input, and switch to the next inspection step. We designed a user interface for this task, with the aim of achieving similar handling in different interaction modalities without hurting oneself.

For the selection of a faulty weld, we considered direct touch to be the best solution. Pointing directly at the workpiece is fast, contains 3D information, and does not require a cognitive load, e.g., for transformation on 2D media or travel time to a separate interface. We used numbers to code the different defects (1–4) and colors (red, orange, green) as proxy for the handling mode.

This workpiece was a metal frame in a dimension of 100 cm by 60 cm and its center was 140 cm above the floor and in an 80 cm distance from the user. To adjust the system to the height of the test subjects, a step was used (in our setup, we had a fixed workpiece position in the absence of a robot, as can be seen above). In front of the workpiece, there was a worktable which was 100 cm high and 200 cm wide, onto which the worktable interface was projected.

The active regions on the metal frame (buttons and weld) were approximately 5cm×5cm (slightly wider than the frame) and the buttons on the worktable were 22cm×22cm. A frame button was activated by a dwell time of 600 ms and on the worktable by a vertical downward movement with a hysteresis of 2 cm. When the system recognized a potential missing hysteresis on the worktable or the hand did not move out of the interaction area on the workpiece, the user was notified by a voice output. All UI elements were within reach of both hands, and the interaction scheme did not favor one hand over the other.

Our system was setup in a laboratory with constant illumination. The study was conducted in May 2022 using Chrome with Web Speech SDK for speech recognition. The UI language was in German or English according to the test subject’s preference.

### 5.2. Procedure

The experiment was preceded by a short introduction to the system. The test subjects had to complete four conditions of the simulated inspection task in an accurate and fast manner using different interaction modes. In each condition, 30 welds had to be evaluated. The order of the first three unimodal modes “touch on worktable” (see Figure 10), “touch on workpiece” (Figure 11) and “voice control” were randomly chosen. The fourth condition was always the multimodal condition, where there was no restriction on when to use which modality. Before each condition, the experimenter briefly demonstrated the interaction, and then the test persons tried the interaction themselves. Each cycle consisted of the following steps: (1) detect defective weld; (2) select defective weld; (3) categorize defect type; (4) select defect severity; (5) confirm decision (5 actions) ∗ (30 repetitions) ∗ (4 interaction modes) = 600 actions. The test subject received verbal instructions on what to select in each step. Participants were asked to work quickly, but not rushed. After each condition, participants completed the NASA TLX, SUS, Kunin scale, and UEQ-S forms in a web-based questionnaire.

At the end of the trial, demographic data were entered into a web-based form by the participant. The experiment (all four conditions in succession) lasted approximately 45 min in total.

### 5.3. Participants

In total, 22 test subjects (15 males and 7 females, among which 15 were between 20 and 29 years old, 4 were between 30 and 39 years old, 1 was between 40 and 49 years old, and 2 were between 50 and 59 years old), students and staff from our institute, were recruited. No compensation was paid. The height of the participants ranged from 155 cm to 194 cm and no physical handicap was reported. Thirteen participants had no or little experience with augmented reality. To ensure the ethical safety of the user studies reported herein, we followed the self-assessment of the Technical University Berlin ethics committee. The questionnaire for the fast-track procedure [46] did not identify any need for action or a separate application to the ethics committee.

### 5.4. Hypotheses

We expect some significant differences in the objective and subjective measurements between interaction modes. It is expected that the task completion time for voice control will be longer than for touch modes, while the physical demands should be less. In terms of task completion time and user preference, the multimodal mode is expected to be the best as each user can use their preferred mode.

## 6. Results

The results are made up of both objective measurements and various subjective assessments.

### 6.1. Objective Measurements

We distinguished between two types of errors. The first is an incorrect input by the user and the second is a malfunction of our system. As false input, we defined when a user pressed an incorrect button, and as malfunction, we defined interactions longer than 5 s. In pre-trials, we did not note any unintended presses of a button or mistaken speech recognition, but we saw that users hit a button unsuccessfully (e.g., hand beside button, hand retraction before the end of dwell time) or that speech recognition did not recognize any valid phrases. In both cases, the system did not switch to the next step and the user had to repeat the input—so we measured a longer execution time. The average false input rate was below 1%. There was no significant difference between the variants. For hand interactions, the system detected in 1.1% of the cases under the workpiece conditions and in 0.1% of the cases under the multimodal conditions that the user did not observe the hysteresis and gave a verbal hint. It was considered a malfunction if more than 5 s were required for an interaction. In 8.4% of the voice interaction mode cases, speech recognition took more than 5 s. For the workpiece conditions, exceeding 5 s was observed in 6.3% and in 4.9% of the interactions in the table. For the multimodal interaction, we saw such timeouts only in 2.7%. This lower error rate may be due to the fact that multimodal interaction was always the last experimental condition, although no significant sequence effects could be found with regard to the task completion time.

Sphericity for the task completion time is not given. The Huynh–Feldt correction [47] was used, as ϵ>0.075; we adapt the degrees of freedom in our ANOVA correspondingly. The Huynh–Feldt corrected ANOVA shows that the average performance difference between the groups is statistically significant, F (2.539, 53.329) = 5724, *p* = 0.003, partial $\eta^{2}$ = 0.214. In the post hoc procedure, individual comparisons showed significant differences between the multimodal mode and the workpiece mode (MDiff = −107.229, 95%-CI [−193.832, −20.626], *p* < 0.01), and between the multimodal mode and the unimodal speech interaction (MDiff = −106.356, 95%-CI [−151.186, −61.527], *p* < 0.001). In all comparisons, the multimodal mode was superior in terms of interaction time. The remaining differences—worktable and workpiece (*p* = 0.680, MDiff = −61.915 95%-CI −171.059, −47.228]), worktable and speech (*p* = 0.604, MDiff = −61.043 95%-CI [−164.591, 42.505]), and workpiece and speech (*p* = 1, MDiff = −0.873, 95%-CI [−91.157, −89.411])—were not significant.

Only three of the 22 test subjects used a single interaction type in multimodal mode. This was always the interaction on worktable, allowing for the fastest interaction (see Figure 12). On average, speech was used in 24% of cases, the worktable was used in 58% of cases and the workpiece was used in 18% of cases (Figure 13). We did not find any relation between the given task and the choice of modality.

No significant correlation in the objective measurements with body size or any other attribute of a test subject using bootstrapping were found. The biggest effects across all groups were weak to moderate correlations between the variables’ height and worktable (r = −0.229, 95% CI [−0.809, 0.278], *p* = 0.345), arm length and worktable (r = −0.334, 95% CI [−0.697, −0.322], *p* = 0.162).

### 6.2. Subjective Measurements

#### 6.2.1. NASA TLX

NASA TLX shows only small differences (Figure 14 and Table 1). Sphericity is given (*p* = 0.230), and within-subjects effects F (3, 63) = 5.722, *p* = 0.002, partial $\eta^{2}$ = 0.214. In the dimension of the physical demands (F = 17.302 *p* < 0.001 $\eta^{2}$ = 0.452), it was found that the interaction at the workpiece is more demanding than speech (*p* < 0.001) or multimodal interaction (*p* = 0.003). In the dimension of frustration (F = 7.254 *p* = 0.002 $\eta^{2}$ = 0.254), a significant difference can be found between multimodal and speech (*p* = 0.005). For all other dimensions, no significant differences can be found at *p* < 0.005. In the dimension performance and frustration, there is a wide range of user ratings.

#### 6.2.2. System Usability Score—SUS

It can be seen that the worktable variant achieved the highest average SUS score of 86.591 (SD = 10.593), closely followed by the multimodal variant with 85.455 (SD 11.615). The workpiece and speech variants scored less well at 80.795 (SD = 12.733) and 73.636 (SD = 18.204), respectively. To test for normal distribution for repeated measures ANOVA (rmANOVA), both the Kolmogorov–Smirnov test and the Shapiro–Wilk test were added. Because the Shapiro–Wilk test demonstrates greater power, its values are preferred: worktable (*p* = 0.090); workpiece (*p* = 0.074); speech (*p* = 0.033); and multimodal (*p* < 0.001). As the Greenhouse–Geisser epsilon ϵ=0.654 is large and thus less than 0.75, the Greenhouse–Geisser correction is preferred over the Huynh–Feldt correction. Since the result is statistically significant, we can use a post hoc test to see which groups are statistically significantly different from each other. Using Bonferroni-corrected individual comparisons, this showed a significantly (*p* = 0.019) higher performance in the worktable group than in the speech group (MDiff = 12.955, 95%-CI [1.603, 24.306]), and also significantly (*p* = 0.002) better in the multimodal variant than in the speech variant (MDiff = 11.818, 95%-CI [3.630, 20.006]) (see Table 2 and Figure 15).

#### 6.2.3. User Experience Questionnaire—Short (UEQ-S)

Huynh–Feldt correction [47] shows that the average performance of the groups is statistically significantly different, F (2.163, 45.413) = 8.437, *p* < 0.001, partial $\eta^{2}$ = 0.287. According to Cohen’s interpretation, this can be considered a large effect (>0.14) [48]. Table 3 lists the significant results (*p* < 0.05). It turned out that the multimodal conditions have a higher user experience quality (pragmatic, hedonic, and overall) than our speech conditions (Figure 16 and Figure 17).

#### 6.2.4. User Satisfaction

We used the Kunin scale [45] to measure the general satisfaction with the usability of the tested conditions.

The Mauchly-test reveals that sphericity can be assumed (*p* = 0.250), which makes a correction unnecessary. There are very significant differences between the different interaction modes (*p* ≤ 0.001). Therefore, the different variants are considered by means of post hoc tests. Interactions under worktable (*p* = 0.005) and multimodal (*p* = 0.001) conditions are superior to speech mode and interaction on a workpiece is superior to speech (*p* = 0.045) (see Figure 18).

#### 6.2.5. Overall Ranking

At the end of the experiment, after the subjects experienced all four conditions, they were asked to rank them in order of preference (rank 1–rank 4). The question was “Which variant do you prefer?”. In this overall ranking, the multimodal condition performed best, while speech received the lowest rating (Figure 19). No one gave the lowest rank to multimodal interaction. The results of the overall ranking is in accordance with the Kunin scale results.

The multimodal mode exhibits a mean rank of 1.45 and the worktable mode of 2.14. Friedman’s two-way ANOVA by ranks shows that the difference is significant (*p* = 0.047). Multimodal and worktable are better ranked than workpiece (*p* < 0.004). Ranking difference between the workpiece and speech is not significant (*p* = 0.161).

## 7. Discussion

The present investigation is part of a larger research project and focuses on one specific research aspect: the comparison of uni- and multimodal interfaces. The handling of the workpiece by the robot was not the focus of this experiment. In order to make the results independent of a single use case, a static workpiece position was chosen. This is permissible because, in the use case of the research project—as in most human–robot collaborations—interactions only take place when the workpiece is at rest (in changing positions). Distractions caused by the workpiece movements of the robot were avoided by the chosen setup for the test subjects who were untrained in interacting with robots. The results of the present laboratory experiment are therefore likely to be transferable to productive use by skilled personnel who are experienced in interacting with robots.

Neither in objective nor subjective measurements did we find evidence that the body height, arm length, sex, or age of test subjects were factors influencing the interaction performance. While we compensated for the effect of users’ body height by adapting the height of the workpiece, all other parameters were fixed for all users. Thus, user parameters, especially the arm length, did not seem to be an influencing factor in reaching buttons on the workpiece for any of the conditions. To test whether the arm length had an influence over some outcomes, e.g., physical demands in NASA TLX, much longer tests need to be conducted.

It is hypothesized that the interaction on a workpiece causes a higher workload than on a worktable because the arm is in a higher position [49]. We found a statistically significant (*p* = 0.016) difference in the physical demands between the worktable and the workpiece (see Table 1), confirming this hypothesis. Additionally, users did rate speech as the least physically demanding modality. Thus, the greatest difference in this dimension is between the speech and workpiece interaction.

It is surprising that the speech input, as the least preferable among the unimodal conditions, was frequently used under the multimodal conditions. However, as Figure 15 reveals, speech interaction has a high SUS value and thus a high usability. Although the SUS score for speech interaction is significantly smaller than the worktable interaction, it still provides a high level of usability (an SUS score above 80.3 is usually considered excellent, and a score between 68 and 80.3 is considered good).

Even though users show a clear preference for a touch-based system, they took advantage of the speech modality when working with a multimodal system. We concluded that the integration of even less optimal interaction modes in a multimodal system provides complementary benefits to the users.

Similar findings were found in a multimodal experiment study, where the users also rated the speech input as less favorable in comparison to gesture input when selecting objects on a table and giving commands to a robot. The implementation relied on using activation keywords, to start the speech input and to make the system “listen”, which further increased the processing time and the frustration of the user [50]. In [18], it was found that the usage of multimodalities depends on the task, although in their experiments, also all modalities were used.

In contrast to [23], we did not find a suitable workflow for hand gestures. While Rupprecht et al. managed with two simple hand gestures (forward/backward) to control the entire workflow, we would need significantly more different hand gestures to implement the complete workflow. Intuitive, simple and easy to remember gestures for the different commands could not be found, so we did not implement such an interaction modality. Maybe an "ok" gesture can be integrated in our workflow, but the performance of a modality in a single step is difficult to compare to more frequently used ones.

### 7.1. Limitations

Our participants did not have any experience with our inspection use case, thus, the actual visual inspection task was intentionally not part of our evaluation. Our trials lasted only 45 min instead of the usual two-hour shifts in the real workplace. However, we focused on the interaction modalities and assumed that our results are transferable. However, we cannot take into account the user’s adaptation to the task and input modalities, which may affect usability.

We optimized our system for an optimal workpiece position during the inspection task. However, the physical demands of inspection and interaction are different: for inspection, a high position in front of the user’s eyes is more ergonomic, while for interaction, a lower interface at hip height is more suitable. The same difference holds workpiece versus worktable interaction modes. Our study might be biased by the influence of the different interaction spaces.

The used camera technology and the projective interface restrict our system to indoor environments only. In bright sunlight, the implemented Microsoft Kinect camera would not work and the projection would be invisible.

### 7.2. Future Work

It would be greatly appreciated to see whether user preferences change when the system is used over a longer period of time. Furthermore, it would be interesting to evaluate whether the chosen workpiece position is optimal or whether real users might prefer a compromise between a comfortable inspection height and the effortless marking of weld seams.

In user trials, we examined only a few functionalities of our system. The integration of robotic functions and the gesture recognition were not used. In future research, a version of the system with a real workpiece, some hand gestures and voice input in a real industrial environment should be tested.

## 8. Conclusions

We showed that multimodal interaction is preferred over a single modality. This confirms previous findings for spatial HRI [18,51]. It is not deducible from objective measurements in single modalities which mode will be used when all modalities are offered at once. Spatial interaction such as touch on an augmented workpiece interface is not favored over touch on an augmented worktable, but users still use it when offered in a multimodal interaction scenario. When evaluating different interaction modalities for a certain task, it is advisable to implement a true multimodal one as well.

In our experiments, we showed that common myths are not true. Myth #4 from [2] cannot be confirmed. Speech was used in our experiments when the user could use it in multimodal mode, but it was not dominant. We can confirm the findings in [18] that multimodal interfaces are preferable.

## Figures and Tables

**Figure 1 sensors-23-05043-f001:**
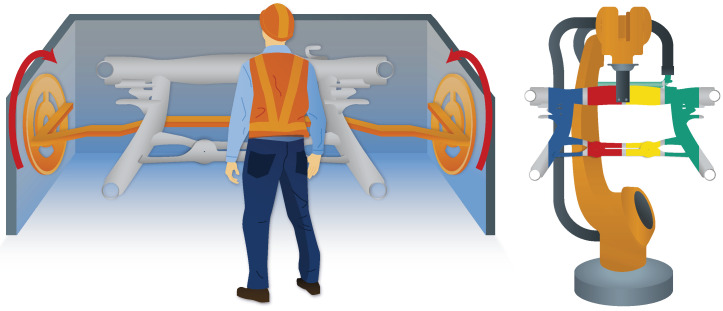
The current approach (**left**) involves manual position changes in fixed orientations without the ability to digitally document defects. The proposed robot-assisted visual inspection station (**right**) presents the component at eye level in arbitrary poses and orientations.

**Figure 2 sensors-23-05043-f002:**
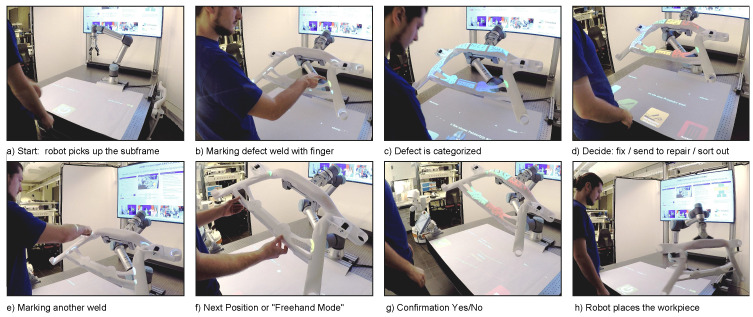
The inspection process of a car subframe. Here, a mockup piece is used. The user interacts directly with the work environment via hand and voice commands. A projected interface is displayed on the table surface and the 3D object. It moves with the object due to real-time tracking.

**Figure 3 sensors-23-05043-f003:**
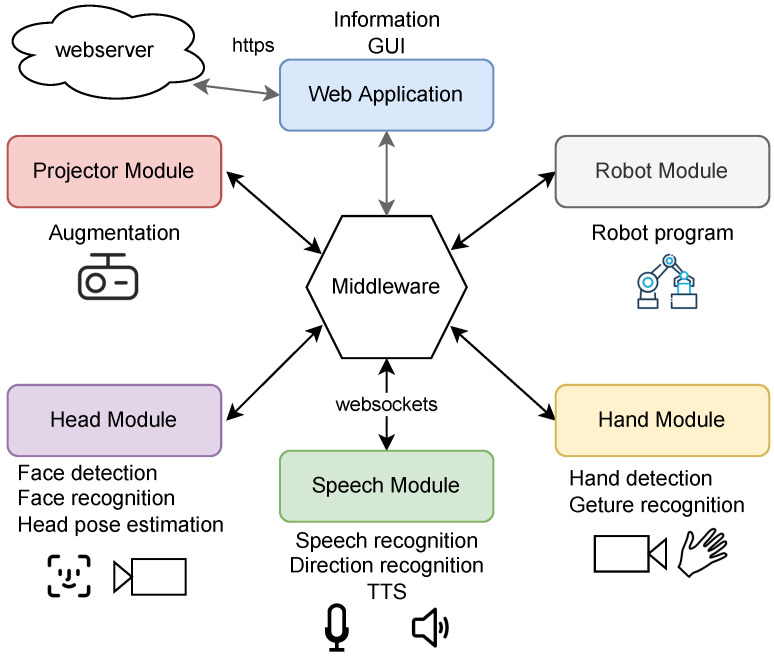
Overview of the system structure and module communication.

**Figure 4 sensors-23-05043-f004:**
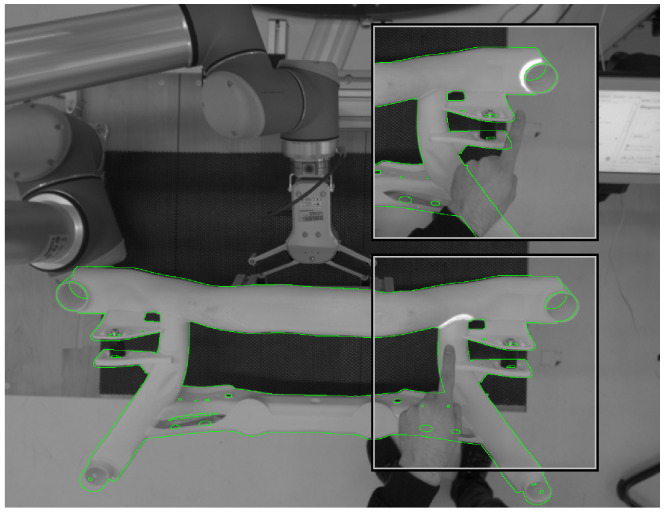
Visualization of the weld selection process as seen by the projector–camera system with overlay (green line) of the contour of the registered 3D model.

**Figure 5 sensors-23-05043-f005:**
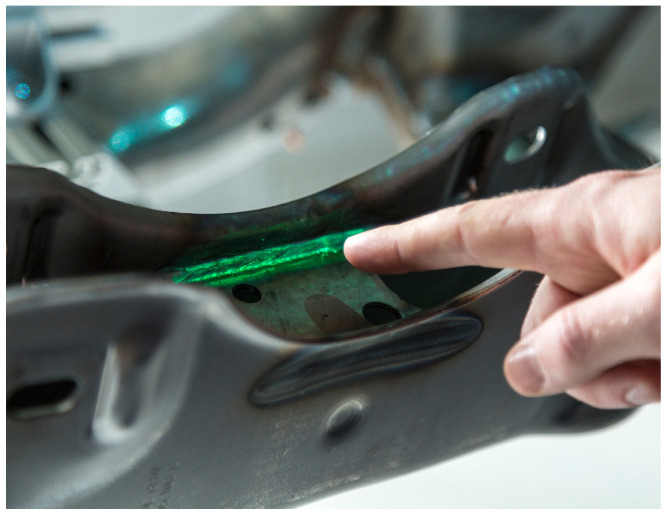
The projector highlights a weld on the original metal workpiece.

**Figure 6 sensors-23-05043-f006:**
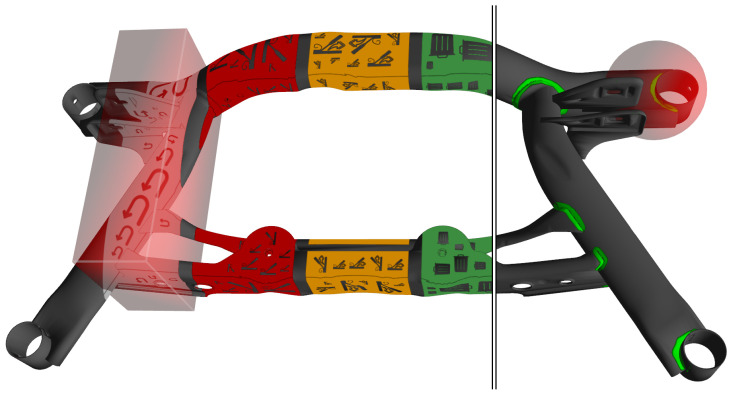
Weld annotations (**right**) and menu items (**left**) are modeled as texture overlays. Hand interaction areas (shown in red for two examples) are modeled as cubic or spherical primitives registered on the 3D mesh.

**Figure 7 sensors-23-05043-f007:**
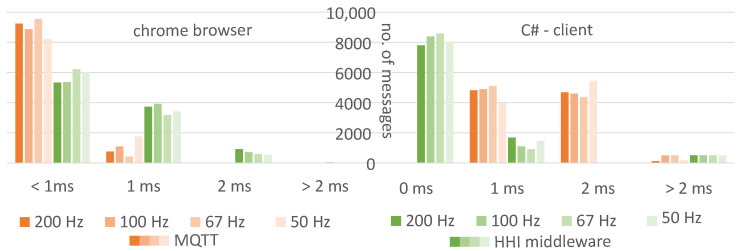
Latency comparison of MQTT vs. our middleware for four sender frequencies (200 Hz, 100 Hz, 67 Hz, 50 Hz) received in a browser (**left**) or a C# client (**right**).

**Figure 8 sensors-23-05043-f008:**
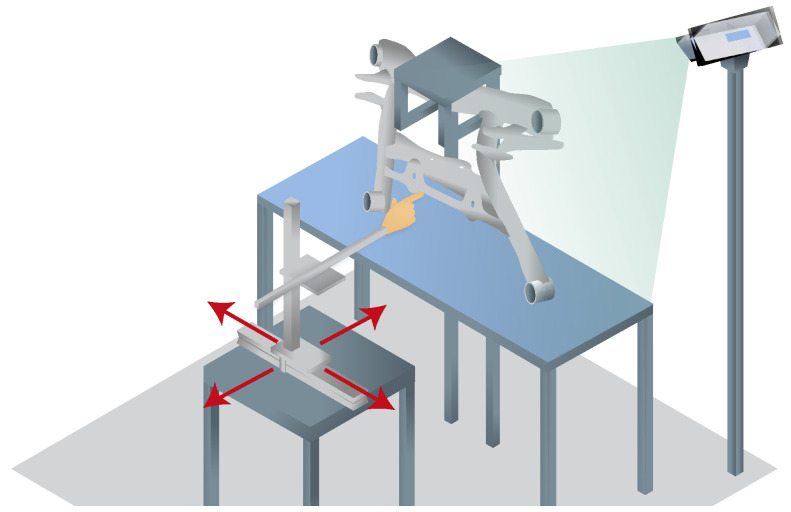
Measuring setup with an artificial hand and measuring table.

**Figure 9 sensors-23-05043-f009:**
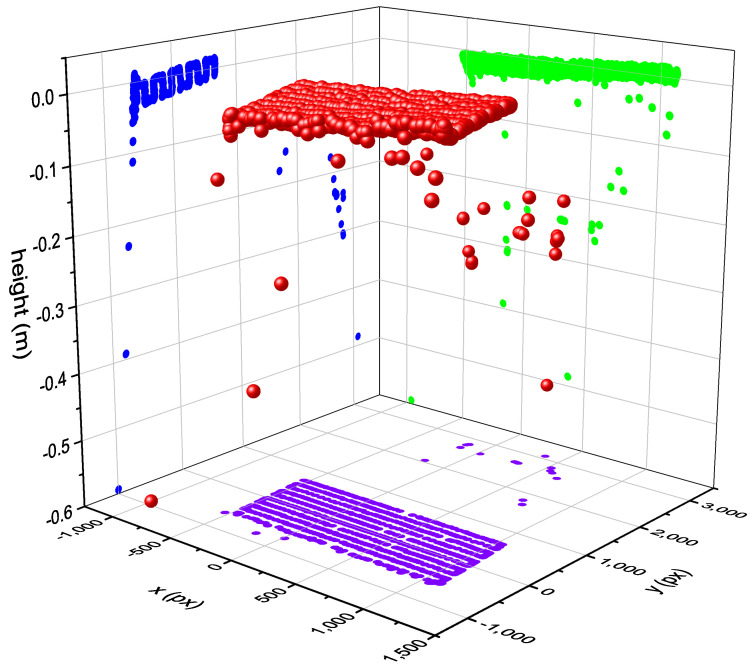
Transformed tracking data with attention to the critical height measurements of the hand. The hand was moved in parallel to the table. The red spheres in the diagram represent the 3D measurements. In purple, blue and green, the values in each dimension are shown on the respective diagram plane.

**Figure 10 sensors-23-05043-f010:**
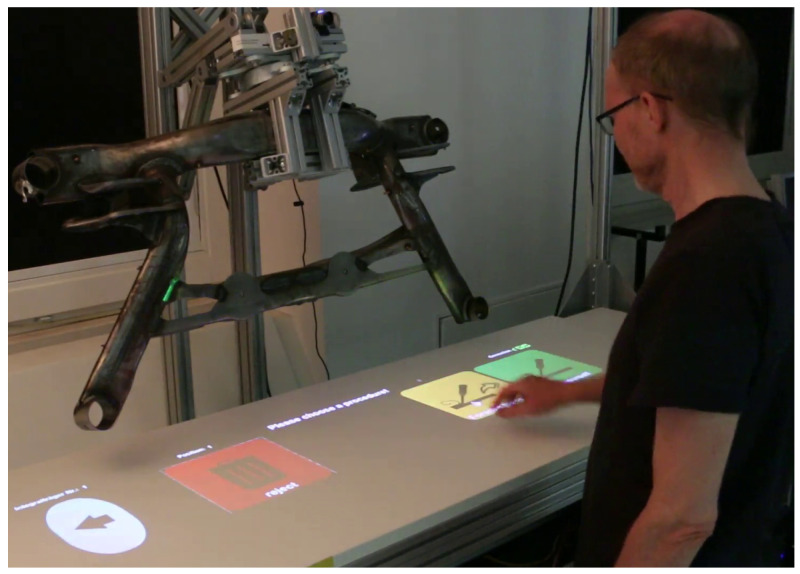
WORKTABLE user interface.

**Figure 11 sensors-23-05043-f011:**
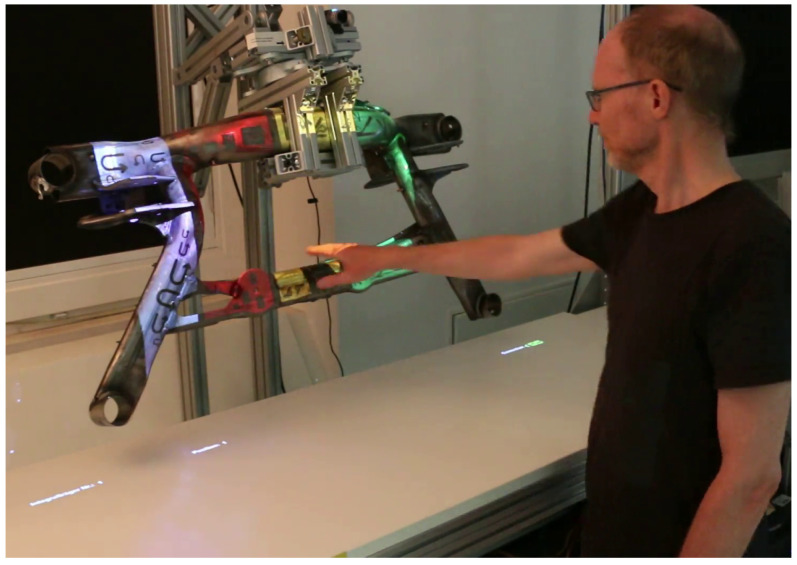
WORKPIECE user interface.

**Figure 12 sensors-23-05043-f012:**
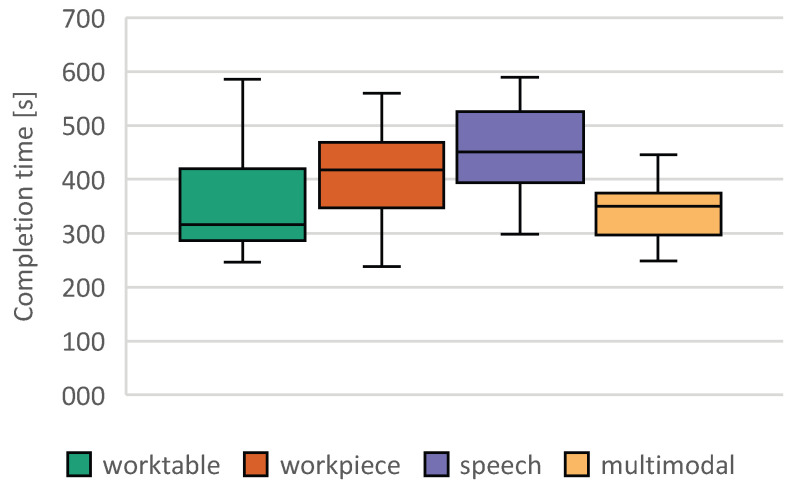
Mean task completion time under each test condition (error bars denote 95% CI).

**Figure 13 sensors-23-05043-f013:**
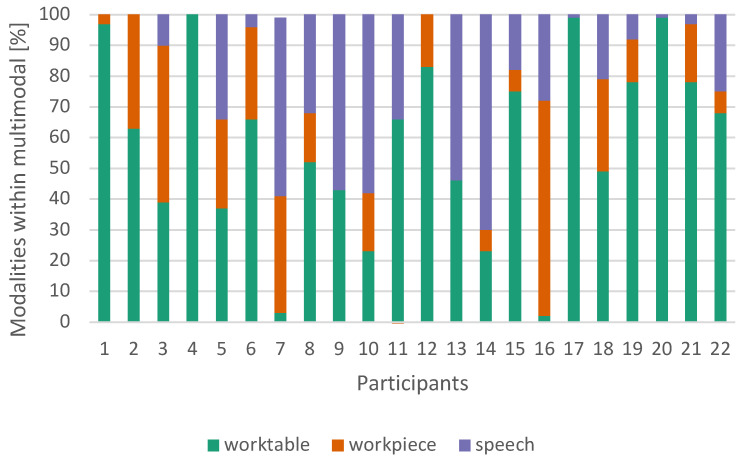
Subjective interaction preferences of each participant under the multimodal conditions.

**Figure 14 sensors-23-05043-f014:**
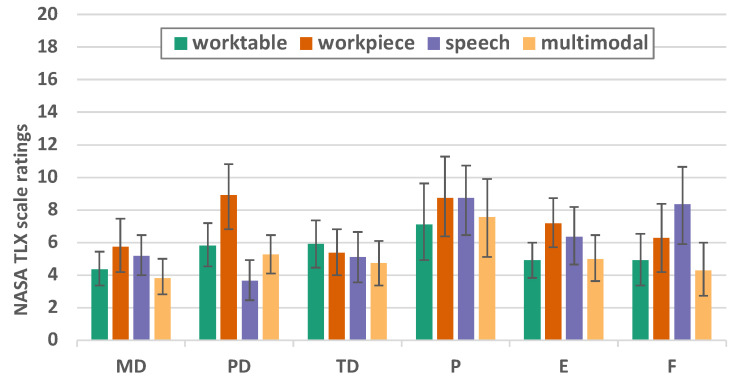
NASA TLX factor values for mental demands (MDs), physical demands (PDs), temporal demands (TDs), performance (P), effort (E), and frustration (F) (error bars denote 95% CI).

**Figure 15 sensors-23-05043-f015:**
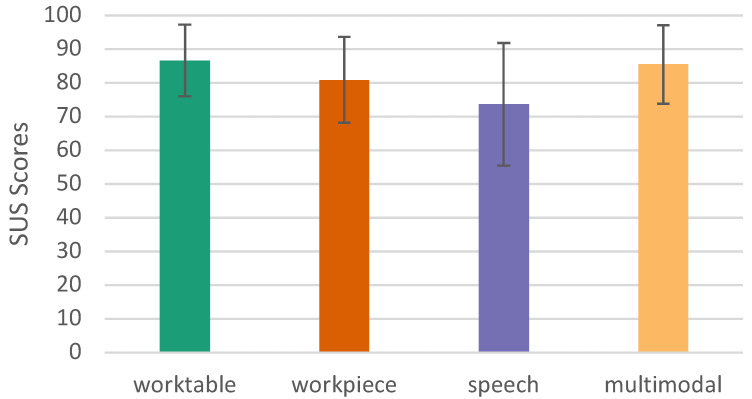
System Usability Score (SUS), error bars show 95% CI.

**Figure 16 sensors-23-05043-f016:**
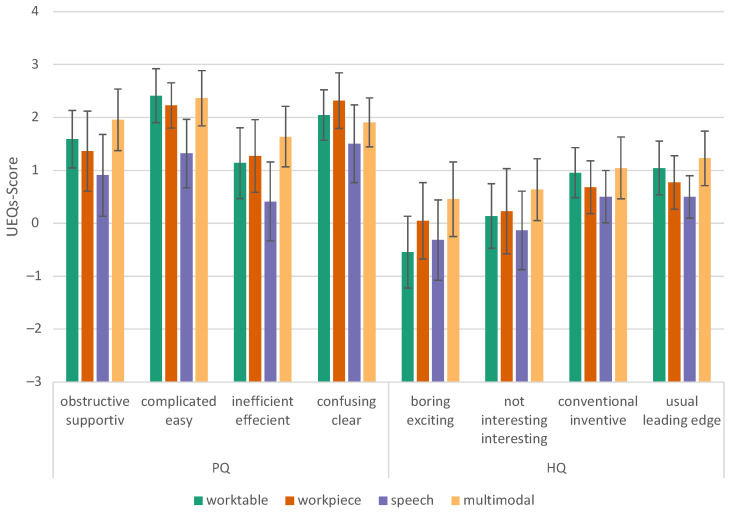
User Experience Questionnaire, components view, error bars show 95% CI, PQ = pragmatic quality; and HQ = hedonic quality.

**Figure 17 sensors-23-05043-f017:**
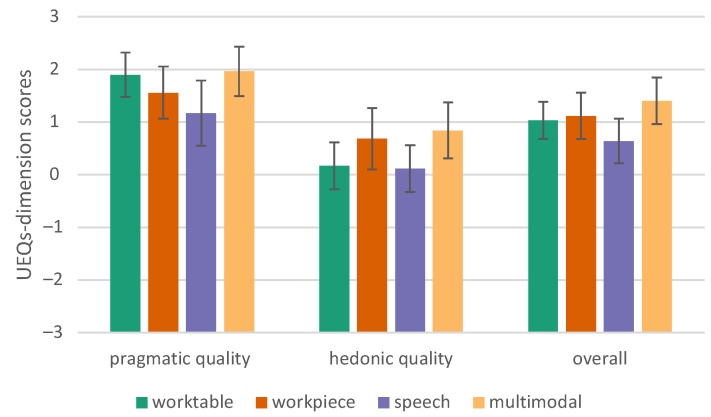
User Experience Questionnaire—short, error bars show 95% CI.

**Figure 18 sensors-23-05043-f018:**
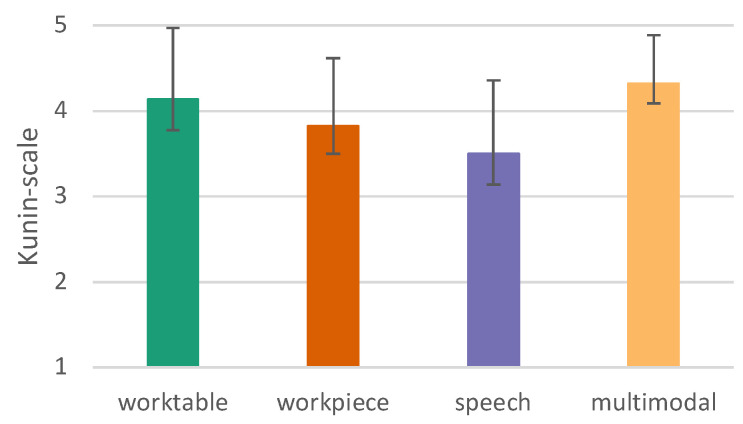
User satisfaction measured by the Kunin scale (1–5, 5 = highest score), error bars show 95% CI.

**Figure 19 sensors-23-05043-f019:**
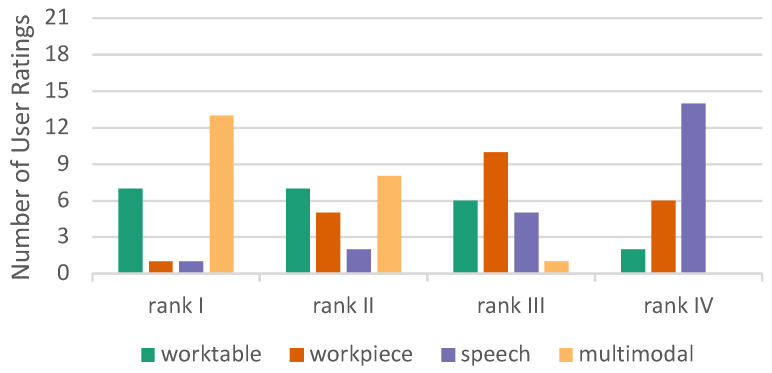
Preferences of interaction modes.

**Table 1 sensors-23-05043-t001:** Pairwise comparison of NASA TLX components, physical demands (PDS), effort (E), and frustration (F).

						95%-CI	
	(I) Group	(J) Group	MDiff (I-J)	SE	*p*	Lower	Upper
**PD**	worktable	workpiece	−3.091	0.909	0.016	−5.738	−0.444
	speech	worktable	−2.182	0.683	0.026	−4.170	−0.194
		workpiece *	−5.273	0.917	<0.001	−7.943	−2.603
		multimodal	−1.636	0.468	0.013	−2.999	−0.274
	multimodal	worktable	−0.545	0.478	1.000	−1.937	0.846
		workpiece *	−3.636	0.889	0.003	−6.225	−1.048
**E**	worktable	workpiece	−2.273	0.724	0.030	−4.382	−0.164
		speech	−1.455	1.029	1.000	−4.452	1.543
		multimodal	−0.091	0.664	1.000	−2.025	1.843
	speech	workpiece	−0.818	0.918	1.000	−3.491	1.855
	multimodal	workpiece	−2.182	0.755	0.053	−4.380	0.016
		speech	−1.364	0.725	0.444	−3.476	0.749
**F**	worktable	workpiece	−1.364	0.794	0.603	−3.675	0.948
		speech	−3.455	1.193	0.052	−6.928	0.019
	workpiece	speech	−2.091	1.143	0.490	−5.420	1.238
	multimodal	worktable	−0.636	0.533	1.000	−2.188	0.915
		workpiece	−2.000	0.811	0.134	−4.362	0.362
		speech	−4.091	1.057	0.005	−7.168	−1.014

* significant.

**Table 2 sensors-23-05043-t002:** Pairwise comparison SUS.

					95%-CI	
(I) Group	(J) Group	MDiff (I-J)	SE	*p*	Lower	Upper
worktable	workpiece	5.795	2.363	0.138	−1.086	12.677
	speech	12.955	3.898	0.019	1.603	24.306
	multimodal	1.136	1.930	1.000	−4.485	6.758
workpiece	speech	7.159	3.352	0.268	−2.602	16.921
	multimodal	−4.659	2.363	0.371	−11.539	2.221
speech *	multimodal	−11.818	2.812	0.002	−20.006	−3.630

* significant.

**Table 3 sensors-23-05043-t003:** Pairwise comparison of significant UEQ-S results, PQ = pragmatic quality, HQ = hedonic quality, OA = overall quality.

						95%-CI	
	(I) Group	(J) Group	MDiff (I-J)	SE	*p*	Lower	Upper
**PQ**	multimodal	speech	0.795	0.212	0.007	0.177	1.414
**HQ**	multimodal	workpiece	0.670	0.200	0.018	0.087	1.254
		speech	0.727	0.243	0.041	0.02	1.434
**OA**	multimodal	speech	0.761	0.170	0.001	0.268	1.255

## Data Availability

Not applicable.

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
