# Peer review of "Assessing the Value of Multimodal Interfaces: A Study on Human–Machine Interaction in Weld Inspection Workstations"

_sensors, 2023, doi:10.3390/s23115043_

Round 1

Reviewer 1 Report

The article discusses a study on multimodal user interfaces for human-machine interaction in an industrial use case of a weld seam inspection workstation. The study compares three independent interfaces and their combination. The results show that while users preferred the interaction on the augmented worktable when comparing single modalities, they used all three in the multimodal user interface. The authors conclude that the effort of implementing and using multiple input modalities is reasonable.

This article is a significant contribution to the field. The authors have conducted thorough research and presented their findings in a clear and coherent manner.

Author Response

Thank you for your review and the positive feedback. We have tried to revise the text to present our results more clearly. 

Reviewer 2 Report

The authors compared and evaluated four interfaces which are touch interaction on the worktable, on the workpiece, voice control, and multimodal interaction about the human-robot welding seams inspection work. The authors compared the task completion times and the error rate as objective measurements of interface effectiveness.  In all comparisons, the multimodal mode was superior in terms of interaction time. In addition, the authors examined the subjective measurements of each interface mode based on four indices: NASA TLX, System Usability Score, User Experience Questionnaire, and User Satisfaction. In an overall ranking the multimodal condition performed best, while speech got the lowest rating. The reviewer considers that the results of this evaluation are important for developing an easy-to-use and human-friendly interface.

The reviewer has several questions.

1.p.13 Line 432

  Only four of the 22 test subjects used a single interaction type in multimodal mode.

    From Figure 13, the number of test subjects using a single interaction maybe three. (No.4,17,20)

2. p.17  Figure 19

     There is no explanation as to the meaning of rank I through IV in Figure 19.

  Do these mean NASA TXT or SUS?

3.p.18 Line 509-510

   “Although the SUS score for speech interaction is significantly smaller than the worktable interaction, it still provides a high level of usability.”

  How high should the SUS score be considered?  Please indicate the criteria.

Author Response

Thank you for your review and the valuable and positive feedback. 

Please find the answers to your questions below each question:

1. p.13 Line 432

  “Only four of the 22 test subjects used a single interaction type in multimodal mode.”

    From Figure 13, the number of test subjects using a single interaction maybe three. (No.4,17,20)

>> has been corrected to: “Only three….”. We originally considered user 1 also as a unimodal one, since he/she used one modality for more than 95% of the time.

2. p.17  Figure 19

     There is no explanation as to the meaning of rank I through IV in Figure 19.

  Do these mean NASA TXT or SUS?

>> Thank you for this comment, which will help us to make the paper better understandable. Fig. 19 shows a personal preference ranking of each user accessed at the end of the experiment. We clarified that at the beginning of Chapter 6.2.5., before Fig 19.

3. p.18 Line 509-510

   “Although the SUS score for speech interaction is significantly smaller than the worktable interaction, it still provides a high level of usability.”

  How high should the SUS score be considered? ã€€Please indicate the criteria.

>> Thank you also for this comment. We added a clarification on the to interpretation of the  SUS at this point saying that “a SUS score above 80.3 is usually considered excellent, and a score between 68 and 80.3 is considered good.

Reviewer 3 Report

Dear Authors.

I would like to congrats you for the quality of the work presented.

I was in an area that is hard to bring some quantitative data, but with a good literature review, you could bring solid results.

If I could suggest something to improve the quality of your paper is in section 4.1 where you have evaluated the performance of the middleware. What I suggest is to have a comparison with the same time of sampling, but for me, the more accuracy will be the same quantity of samples.

And one more thing, that I could detect if I could say this, is how you have made the tests with persons from a research institute, this could lead to an acceptance better for the technology proposed. But if we go to a company that has workers that don't have this empathy with technology, maybe we could get a different result.

This comment doesn't invalidate all your work, but I think that shows that in a context we have excellent improvement, but in an industrial context the results should be different.

Author Response

Thank you for your review and the positive feedback. Please find our thoughts on your two valid suggestions below.

Regarding your kind suggestion about section 4.1 where we have evaluated the performance of the middleware:

>> Thank you for this valid suggestion.  We have chosen this presentation from our prior experience and practical implications. At the moment we unfortunately cannot repeat the measurements, but we will consider your suggestion for future research.

Regarding your comment about the test subjects:

>> This is an excellent observation. You are right, since our test subjects were mostly persons from a research institute, these results may not be 100% transferable to the industrial context. To provide a higher generalizability we designed the evaluated task very generic and without the need of industry specific knowledge. We wanted to evaluate the modalities in this specific task (marking welds, controlling machine) and in this specific ergonomic set-up (standing person, workpiece in front of the users’ eyes). We know from our industrial partners, which motivated this use case and research, that currently the solution for this task in the car manufacturing is much worse for the employees in terms of ergonomics. Thus, the ergonomic advantage of our solution may have an impact on the evaluation of the input modalities. If tested with experienced industrial inspection workers, the results of the human-machine interaction may be mediated by the fact, that our solution is much more ergonomic for them and thus everything may be rated higher. We believe that this is a general problem of testing concepts in the lab vs. in the field and of course having access to the most representative test subjects.

Reviewer 4 Report

Paper: Comparison of Uni- and Multimodal Interfaces for Spatial Human-Robot Interaction

This paper comparing the uni- and multimodal interfaces for spatial human-robot interaction. I recommend the authors to care with the writing and do all of the following comments to improve the quality of their paper.

Comments:

P1: The abstract is superficial. Deep information about the method and the results should be added.

P2: The main contribution and the importance of this work should be clear in the text of the paper.

P3: In literature review, the advantages and disadvantages of each method should be discussed.

P4: It is not good to start the section with a figure, as the authors did in section 3. Don’t put a figure in the beginning of a section immediately. Also, start any section with a short paragraph describing its content.

P5: The English of the paper should be revised carefully, as there are many errors. In addition, I recommend the authors to reduce using “we”.

P6: It is recommended to add a link in the paper for a video illustrating the executed experiments.

P7: The following recent paper is very related to the topic of the paper. I recommend the authors to read and use it. https://doi.org/10.3390/machines10070591

P8: Comparing the results with other previous published works is recommended to be added in section 7. Discussion.

P9: The conclusion should be rewritten to be in the suitable form.

The English of the paper should be revised carefully, as there are many errors. In addition, I recommend the authors to reduce using “we”.

Author Response

Thank you for your time, the review and the valuable feedback. We have tried to address all your remarks as explained below.

P1: The abstract is superficial. Deep information about the method and the results should be added.

>> We thank you for this suggestion. We have completely revised the title and the abstract.

P2: The main contribution and the importance of this work should be clear in the text of the paper.

>> We thank you for this suggestion. We have tried to revise this topic in the manuscript as much as possible in the given time and also consider the assessment of the other reviewers.

P3: In literature review, the advantages and disadvantages of each method should be discussed.

>> We thank you for this suggestion. We have tried to revise this topic in the manuscript as much as possible in the given time and also consider the assessment of the other reviewers.

P4: It is not good to start the section with a figure, as the authors did in section 3. Don’t put a figure in the beginning of a section immediately. Also, start any section with a short paragraph describing its content.

>> We thank you for this suggestion. We have addressed this topic, Fig. 3 is not at the beginning of section 3 anymore.

P5: The English of the paper should be revised carefully, as there are many errors. In addition, I recommend the authors to reduce using “we”.

>> We thank you for this observation. We have revised the language in several parts of the manuscript and as much as possible in the given time. In line with other recommendations for scientific publications, we tend to avoid the passive voice as much as possible. Unfortunately, this leads to the use of the personal pronoun "we" in the active voice.

P6: It is recommended to add a link in the paper for a video illustrating the executed experiments.

>> A video is added as supplemental material.

P7: The following recent paper is very related to the topic of the paper. I recommend the authors to read and use it. https://doi.org/10.3390/machines10070591

>> We thank you for this suggestion. We added the topic security, which is mentioned in the suggested paper, and was not addressed in detail in our paper before.

P8: Comparing the results with other previous published works is recommended to be added in section 7. Discussion.

>> We thank you for this suggestion. We added the topic security, which is mentioned in the suggested paper, and was not addressed in detail in our paper before.

P9: The conclusion should be rewritten to be in the suitable form.

>> We thank you for this suggestion. We have tried to revise this topic in the manuscript as much as possible in the given time and also consider the assessment of the other reviewers.